# Guinea Worm Disease: A Neglected Diseases on the Verge of Eradication

**DOI:** 10.3390/tropicalmed7110366

**Published:** 2022-11-10

**Authors:** Carmen Pellegrino, Giulia Patti, Michele Camporeale, Alessandra Belati, Roberta Novara, Roberta Papagni, Luisa Frallonardo, Lucia Diella, Giacomo Guido, Elda De Vita, Valentina Totaro, Francesco Vladimiro Segala, Nicola Veronese, Sergio Cotugno, Davide Fiore Bavaro, Giovanni Putoto, Nazario Bevilacqua, Chiara Castellani, Emanuele Nicastri, Annalisa Saracino, Francesco Di Gennaro

**Affiliations:** 1Department of Biomedical Sciences and Human Oncology, Clinic of Infectious Diseases, University of Bari, 70121 Bari, Italy; 2Department of Internal Medicine and Geriatrics, University of Palermo Geriatric Unit, 90121 Palermo, Italy; 3Operational Research Unit, Doctors with Africa CUAMM, 35100 Padua, Italy; 4National Institute for Infectious Diseases Lazzaro Spallanzani-IRCCS, 00042 Rome, Italy; 5Universite du Kwango—UNIK, Kenge, Democratic Republic of the Congo

**Keywords:** dracunculiasis, Guinea worm disease, NTDs, Africa, Spirurida, neglected diseases

## Abstract

Background: Dracunculiasis, also known as Guinea worm disease (GWD), is a neglected tropical disease (NTD) caused by a parasite (*Dracunculus medinensis*). In the past, dracunculiasis was known as “the disease of the empty granary” because of the difficulties patients had in going to work in fields or to school when affected by this disease. In tropical areas, the condition has been widespread in economically disadvantaged communities, and has been associated with reduced economic status and low levels of education. Methods: we searched PubMed, Scopus, Google Scholar, EMBASE, Cochrane Library, and WHO websites for literature addressing dracunculiasis published in the last 50 years. Results: by development and optimization of multi-layered control measures, transmission by the vector has been interrupted, but there are foci in several African countries with a high risk of compromising the results obtained in the control of this neglected disease. Conclusion: this review features state-of-the-art data on the infection prevalence, geographical distribution, diagnostics, parasite–host interactions, and the pathology of dracunculiasis. Also described are the current state and future perspectives for vector control and elimination strategies.

## 1. Introduction

Dracunculiasis, also known as Guinea worm disease (GWD), is a neglected tropical disease (NTD) caused by a parasite (*Dracunculus medinensis*). The name of the disease comes from its prevalence in the Gulf of Guinea. In the past, dracunculiasis was known as “the disease of the empty granary” because of the difficulties patients faced in going to work in fields or to school when affected by this disease [1,2]. *Dracunculus medinensis* (DM) is a nematode and belongs to the order of Spirurida, tissue parasites that produce eggs containing larvae and spread free larvae in the water. The life cycle of this parasite requires arthropods as intermediate hosts. GWD is listed among the common filariases, and it has been widespread in economically disadvantaged communities in tropical regions, including Africa and South Asia, where it has been associated with reduced economic status and low levels of education [3]. Dracunculiasis was targeted for eradication several decades ago, because of its limited geographical distribution, predictable seasonality, and straightforward diagnosis [4]. In the past, it was thought that humans were the unique host, but recent studies have shown that larvae can also infect dogs, which complicates the success of eradication within the time frame foreseen by the Guinea Worm Eradication Program (GWEP) [5]. In this review, we aim to summarize the current knowledge on epidemiology, clinical manifestation, transmission, life cycle, management, and control strategies. Table 1 shows the parasite species, their geographical distribution, and their location within the human host. 

## 2. Materials and Methods

We searched PubMed, Scopus, Google Scholar, EMBASE, Cochrane Library, and WHO websites (http://www.who.int (accessed on 2 October 2022) for literature addressing dracunculiasis, published in the last 50 years. We searched for the literature using the following search strategy: “Dracunculiasis [tiab] OR Guinea worm disease [mh] OR Dracunculus medinensisis [tiab] OR Spirurida, [tiab] OR GWD [tiab] OR water flies [tiab] OR Copepods [tiab])”. All studies dealing with epidemiology, physiopathology, clinical characteristics, screening and diagnosis, therapy, management and eradication programs were included. 

## 3. Epidemiology

The disease is typical of rural communities in low-income countries, whose survival depends on the presence of open surface water. For this reason, disease prevalence highly depends on rain patterns and climate. In arid areas, transmission occurs mainly in the rainy season, when surface water is more easily available [6]. In wet areas, on the other hand, the disease strikes more intensely in the dry season, when drinking water sources are few, as stagnant water collection points, such as wells and cisterns, are well-known parasite reservoirs [7].

GWD can affect people of all ages but is more common in young adults aged 15 to 45, with no difference in prevalence between males and females. In the 1940s, an estimated 48 million people were affected by GWD in Africa, the Middle East, and India, while in the 1980s, only 3.5 million cases per year were reported in 20 countries worldwide, including 17 in Africa. In the same period, the GWEP was initiated, and this led to a sharp reduction in cases in the following years. The number of reported cases dropped below 10,000 for the first time in 2007, falling further to 542 cases in 2012, 54 cases in 2019, and 27 cases in 2020 (of these cases, 1 in Angola, 12 in Chad, 11 in Ethiopia, 1 in Mali, 1 in South Sudan, and 1 in Cameroon) [8]. As of 30 October, according to the WHO, 198 countries, territories and areas have been certified free from dracunculiasis transmission. Seven countries remain to be certified, of which one (the Democratic Republic of the Congo) has no recent history of dracunculiasis [9]. The six other countries are either endemic (Angola, Chad, Ethiopia, South Sudan, and Mali) or are in the pre-certification phase (Sudan). According to WHO and CDC data, 15 Guinea worm cases were reported in 2021 [8,9]. Figure 1 shows the GWD human cases in African countries in 2021.

## 4. Transmission and Life Cycle

Humans contract GWD by drinking unfiltered water from wells and other stagnant water sources infested with copepods, “water fleas” too small to be seen with the naked eye. Guinea worm larvae are ingested by copepods which are ingested, in turn, by people who drink the contaminated water. In the human host, the copepod is dissolved by gastric juice, releasing Guinea worm infective larvae [4,5]. Once in the stomach, the larvae penetrate the gastric or intestinal mucosa and, after a period in the abdominal cavity, migrate up to the connective tissue. Here, *Dracunculus* larvae mature to adult worms and, after mating, the male worms die, and female worms mature and acquire a cylindrical, white and smooth body. The tip of the tail is pointed forming a blunt hook. Female worms take 9 to 14 months to reach adult form, and can measure up to 1 m in length. Approximately one year after infection, female worms migrate through the subcutaneous tissue, reach the surface and emerge from the skin, often at the lower extremities. Before they emerge, they create a painful and itchy blister on the skin site. Affected individuals normally seek pain relief by dipping the lower limbs in cold water, an action that leads to the rupture of the blister and the emergence of the worm from the skin. In this phase, the first stage *larvae* are released into the water and are then ingested by the copepods, reaching the final stage and re-starting their life cycle [10].

Alternatively, infection can occur in humans and dogs after eating aquatic animals, such as fish and frogs, that have ingested infected copepods and may be carriers of Guinea worm *larvae* without presenting signs of infection [11]. If these animals are, in turn, eaten raw or undercooked, they can release the larvae into the digestive tract of the second predator. This alternate cycle has made eradication of GWD more difficult, particularly in Chad [12,13]. The guinea worm life cycle, adapted from the Centers for Disease Control and Prevention (CDC), is illustrated in Figure 2.

## 5. Clinical Manifestations

Usually, patients remain asymptomatic for about a year after infection or may experience aspecific symptoms, such as lymphadenitis or hepatomegaly, due to the growth of non-specific granulomas in the liver and lymph nodes resulting from hypersensitivity or foreign-body reactions to dead male worms [14,15,16]. Concurrently, mature female worms reach the skin, causing a painful papule in the dermis due to the host reaction. Blisters are accompanied by redness and induration and may be preceded by systemic symptoms such as fever, urticarial rash, intense itching, nausea, vomiting, diarrhea and dizziness [15]. More than 90% of worms emerge below the knees, but they can emerge from anywhere in the body. Other common areas of worm surfacing are the head, arms, breasts, back, and scrotum. Usually, infected patients experience one worm emergence per year, but up to 20 (or more) worms may appear at the same time in one individual, and this painful process might last for more than 8 weeks [16].

Generally, patients obtain symptom relief by soaking the affected limb in water or by pouring water over the lesion, which accelerates the sloughing-off of the skin over the blisters [17]. After vesicle rupture, pain and systemic symptoms reduce. Female worms protrude their anterior end from the ulcer and discharge first-stage larvae (640 × 23 µm) into the water. It remains protruding for the following 2–6 weeks, releasing larvae each time the infected body part is immersed in water. After this period, the worm dies [9,10,11]. Worm protrusion is often exacerbated by secondary bacterial infections. In fact, the lack of adequate medical care usually results in superinfections—with an incidence rate that often exceeds 50% of lesions—leading to different complications: local cellulitis, abscesses, tetanus, septic arthritis or sepsis [18]. Tetanus is a serious complication of GWD. Articular manifestations include acute monoarthritis (due to direct invasion of the worm) or arthralgic syndrome secondary to deposition of calcifications in the joints [19]. Joint infection may lead to deformities or contractures, particularly in the knees, where it can evolve into a destructive arthropathy [20]. 

During manual extraction (see below), rupture of the Guinea worm can occur, leading to an intense inflammatory reaction, as the remaining part of the dead worm starts to degrade inside the body. This causes more pain, swelling and cellulitis along the worm tract [15].

During their migration, Guinea worms occasionally end up in ectopic sites such as the pancreas, lungs, periorbital tissue, testicles, pericardium and spinal cord, causing dreadful complications such as pleurisy, pancreatitis, compression of the spinal cord, inguinal adenopathy, compression and abscess formation. In pregnant women, larvae migration may also be responsible for placental bleeding and abortion [21]. Clinical manifestations are summarized in Table 2.

## 6. Diagnosis

Diagnosis of GWD is mainly clinical and it consists of observation of the worm emerging from the blister [15]. Epidemiological considerations have an important role, considering the fact that the blisters cannot be distinguished from other common skin lesions, such as bacterial infections or diabetic foot-related conditions [14]. The diagnosis can be formulated only when the female worm emerges, typically wrapping around a stick. During worm spillage, the diameter of the nematode should be assessed, since bodies that are smaller than 2 mm represent a risk factor for worm body rupture [19]. Active larvae can be obtained by immersing the protruding adult female in a small tube or container with water. First-stage larvae, with their characteristic pointed tails, can then be observed under a microscope. 

Despite the simplicity of diagnosis, GWD can be misdiagnosed due to the initial symptoms’ non-specificity [14]. In addition, in countries where the two parasites are co-endemic, differential diagnosis with *Onchocerca* spp is needed, because of their similar clinical presentation and to inform public health officals in charge of both Guinea worm and onchocerciasis control campaigns. In addition, clinicians should not miss the opportunity to provide correct treatment for *Onchocerca* infections [22,23,24]. Furthermore, when available, species identification allows for better surveillance. Thanks to genomic analysis, the common origin of animal and human infections has been proven [25,26] and new *Dracunculiasis* species are now under study in Vietnam (WHO-defined GWD-free country) [27,28].

Radiological diagnosis is also possible, even if it generally represents an occasional finding. Case reports of breast location are described during mammogram screening [29,30,31]. A typical radiological finding of GWD is the calcification sign. Calcification occurs once the gravid female dies inside the soft tissue. Thus, even when the imaging procedure is performed for pain complaints and not for screening, GWD radiological findings indicate only previous contact with parasites, and no active infection [32]. However, radiography can suggest a relation between systemic symptoms and previous Guinea worm infection. In a case report from Chad, X-ray observation of typical calcifications made clinicians aware of a rare form of past GWD infection related to new onset asthma [33]. In addition, a prospective study correlated localized myalgia (55%), chronic monoarthralgia (35%) and chronic knee synovitis with previous dracunculiasis in view of radiological findings (in two cases, eosinophilia was also present in the synovial fluid) [34].

It is still important to consider that GWD is not the only helminthiasis showing radiological signs. Usually, radiography shows a characteristic long, string-like, serpiginous calcified lesion within the soft tissues. Still, morphological differential diagnosis should rule out other parasite infections: much smaller calcifications located into the hands suggest *Loa loa* and *Onchocerca volvulus*; while multiple “rice grain” calcifications along muscle fibers are indicative of cysticercosis. Generally, these radiological features orientate to an etiological diagnosis. Diagnostic problems can occur when the worm has partially disintegrated, and calcifications are amorphous [32,33,34].

Concerning laboratory values, eosinophil count is often increased [35]. Serological tests can be considered to prove contact with *D. medinensis*. In consideration of possible cross-reactions with other nematodes, ELISA and western blot detection of IgG4 has been observed to obtain the best sensitivity and specificity [35]. A possible confirmation of these findings can be assumed by recent studies by Priest et al. for the realization of multiplex bead assay for seroprevalence assessment among dog populations in endemic areas [36,37]. Nevertheless, serological tests are not included among tools for either seroprevalence assessment or as confirmation tests [38].

## 7. Treatment and Management

First-line treatment of Guinea worm active infection consists of removing the female worm when it comes out of the skin and pulling it out gently to avoid rupture or returning it inside the wound. Worms must be alive during extraction. Usually, a gauze or a small stick is used to allow the worm to roll around it, continuing to exert some traction to bring it out. This is a long process that can take hours or days, because the worm can be longer than a meter. Two actions facilitate the exit of the worm: dipping the affected body part in a bucket with water (to avoid contaminating drinking water) and squeezing the bump to empty the adult worm from the *larvae*, so that it is thinner and can exit from the wound more easily [39,40].

There is no oral anthelmintic medication available for dracunculiasis. Support therapies such as anti-inflammatory drugs and painkillers can be used to reduce edema and pain. Along with frequent dressings with antiseptic solutions, antibiotic ointments may be applied to blisters to avoid wound superinfections. Until the whole worm body has been pulled out, the wound must be covered with medicated gauze and, until successful eradication, an infected person is not allowed to enter drinking water sources [40]. Tetanus vaccination is recommended. GWD-specific vaccines are not available [41]. 

## 8. Eradication Program and Intervention

Dracunculiasis eradication was a goal set out in 1980 in the USA by the Centers for Disease Control and Prevention (CDC) with the GWEP [42]. Eradication can be considered achieved when in one state there are no more cases for at least 12 consecutive months.

In the 1980s, GWD was present in 20 countries, with more than 3 million people infected every year, 90% of whom were living in Africa. There has been a reduction in the overall number of cases, from 3 million to 27 cases at the end of 2020. At this time, the last disease outbreaks were localized in only few sub-Saharan countries. Due to the absence of effective drugs or vaccines, eradication programs are based on simple public health measures [43]. Deadlines for eradication have been progressively shifted from 1991 to 2009, 2015 and 2020. Considering the previously unknown new routes of transmission, the World Health Organization (WHO) has further postponed the target date for the eradication of the disease from 2020 to 2030. Unfortunately, the 2020 goal has been hampered by an increase of dog infections in Chad; the emergence of the first known cases in Angola; infections in baboons in Ethiopia; and conflicts that broke out in Mali, Sudan and South Sudan [44].

In Chad in 2010, after 10 years of no reports, new GWD cases were documented in people who lived along the Chari River [45], with suboptimal surveillance being the likely cause of disease recurrence. Indeed, the country also reported its first cases among dogs in 2012. Dogs became infected by eating aquatic animals. Since then, there have been several reports of infections in animals such as dogs, cats, and baboons. Smaller outbreaks of GWD in dogs have previously been reported in Mali and Ethiopia. The significant number of animals infected with GWD represents a major challenge for global eradication.

The strategies of the global dracunculiasis eradication campaign, a CDC initiative, are based on case reporting (thus identifying all communities with endemic transmission), on water decontamination and hygiene education [46]. Simple filtration systems to purify water are needed to prevent infection spread. Water chlorination or, at home, fine mesh gauze or boiling are useful to kill larvae, making the water safe for home use. In addition, larvicides have been approved to kill the tiny crustaceans infected by the larvae. A population education program was implemented, which recommended cooking food (especially fish) and gave behavior rules in case of infection [47]. The discovery that pets could be a reservoir of infection reinforced the eradication campaign with the offer of a premium to anyone who reported the appearance of infection in their pet, with free dog care and feeding being offered until the worms were removed. Nevertheless, the main challenge remains the maintenance of wells built in recent decades with the aim of providing drinking water, that may often be many kilometers away from the most isolated houses [7]. Further challenges are represented by the need for a constant water supply, difficulty in digging a specific territory, and funding surveillance conservation activities [43].

## 9. Conclusions

Dracunculiasis is a disease that is still present in some low-economic development areas of the world, where the CDC’s global dracunculiasis eradication campaign and the GWEP have been underway for many years [48,49,50,51]. The difficulty in eradication derives from the poor control of the sources of infection and the difficulty in reaching the rural areas where it is still present. Nevertheless, the eradication campaign has achieved excellent results over the last twenty years. Only 14 human cases were reported in 2021 in 14 villages. At the end of 2021, there were 187 countries certified free of dracunculiasis [49,50]. The reduction—nearly a 50% drop from the 27 cases reported in 2020—is the result of a nearly 40-year effort by international organizations and national governments to permanently eliminate GWD. If successful, it will join smallpox and rinderpest as the only diseases to have been completely eradicated in human history [48,49,50,51]. In response to infection, the WHO coordinates and monitors the surveillance in Guinea worm-free areas as well. The ultimate goal is to reach eradication worldwide by 2030. It is also crucial to maintain the high quality of studies and clinical attention on neglected diseases in Europe as well and especially in Italy [52], the gateway to the Mediterranean, which can make an important contribution to the control and definitive eradication of this disease, and other neglected diseases [53].

## Figures and Tables

**Figure 1 tropicalmed-07-00366-f001:**
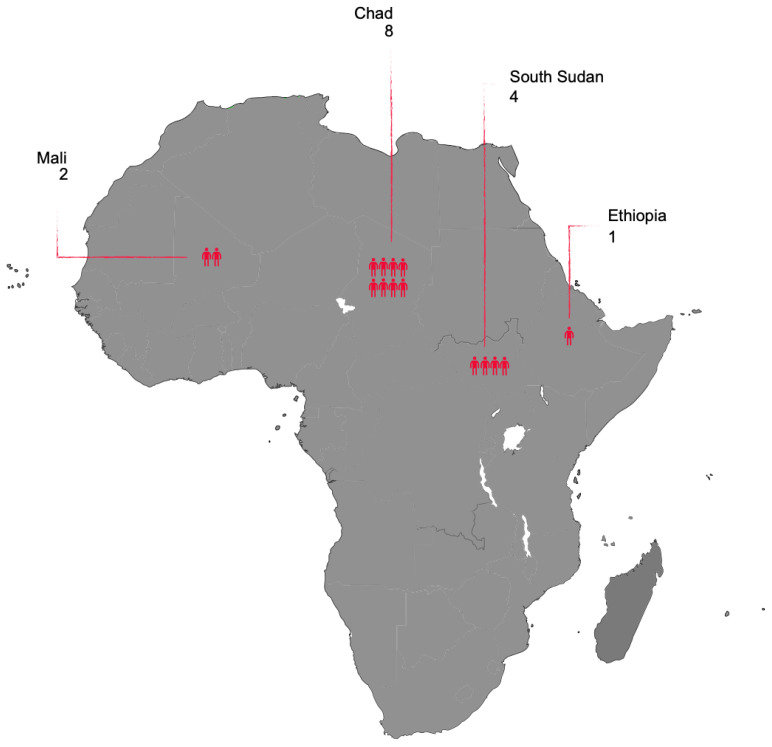
GWD human cases in 2021.

**Figure 2 tropicalmed-07-00366-f002:**
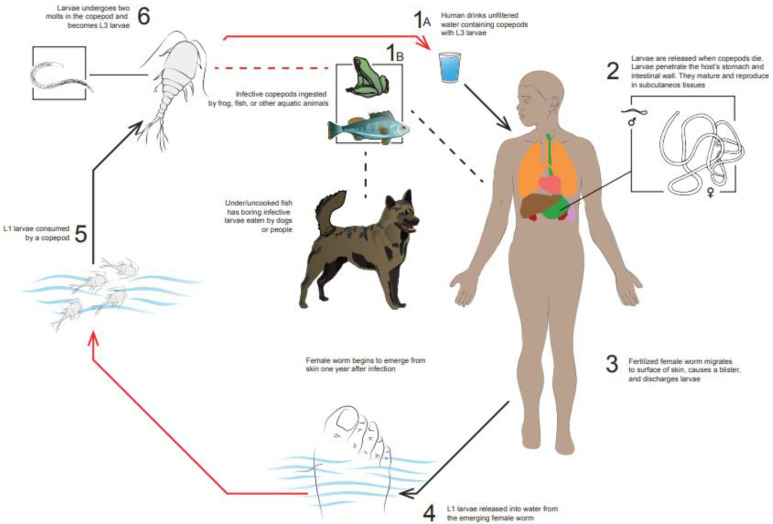
GWD life cycle.

**Table 1 tropicalmed-07-00366-t001:** Filarial parasites and Guinea worm.

Species	Distribution	Vectors	Adult Form Location	Microfilariae Location
*Wuchereria* *bancrofti*	Tropical countries	*Mosquito* spp.	Lymphatics	Blood
*Brugia malayi* and *B. timori*	East Asia (timori in Indonesia)	*Mosquito* spp.	Lymphatics	Blood
*Loa loa*	Central and West Africa	*Chrysops* spp.	Connective tissue	Blood
*Dracunculus medinensis*	Africa and India	Copepods	Connective tissue and Skin	
*Onchocerca* *volvulus*	Central Africa and South America	*Simulium* spp.	Skin	Skin
*Mansonella perstans*	Central Africa and South America	*Culicoides* spp.	Skin	Skin
*Mansonella streptocerca*	Central and West Africa	*Culicoides* spp.	Skin	Skin
*Mansonella ozzardi*	America (Central and South)	*Culicoides* spp.	Serous membranes	Blood and skin

**Table 2 tropicalmed-07-00366-t002:** Clinical manifestations of GWD.

Local	Vesicle, induration, redness
Systemic	Fever, rash, itching, nausea, vomiting, diarrhea, dizziness, hepatomegaly, lymphadenopathy
Joints	Destructive arthropathy
Ectopic sites	Pleurisy, pancreatitis, compression of the spinal cord, inguinal adenopathy, local compression and formation of abscesses, non-specific granulomas
Pregnancy	Placental bleeding, abortion

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
