# Peer review of "Guinea Worm Disease: A Neglected Diseases on the Verge of Eradication"

_tropicalmed, 2022, doi:10.3390/tropicalmed7110366_

Round 1

Reviewer 1 Report (Previous Reviewer 1)

First, I would like to thank you for considering most of my suggestions. I could notice the changes.

I only have a few comments.

Line 187: You should include a period after spp.

Lines 175 – 185:  Remove bold formatting.

Line 285: According to the WHO and CDC (references 8 and 9), there were 15 human cases in 2021.

Line 286: There are two periods after [49-50].

Figure 1: GWD human cases in 2021 instead of 2020.

Figure 1: In 2021, according to reference 9 (WHO) the number of cases were 15 instead of 14. Meanwhile, reference 8 (CDC) reported 14 cases. Please double-check and change the number of cases or the reference in Figure 1 and Line 88.

Author Response

REVIEWER 1

First, I would like to thank you for considering most of my suggestions. I could notice the changes.

 Response: We thank you very much for the encouraging feedback on our manuscript and to give us the opportuntity to improve it. We followed your suggestions and believe that now the paper is more usable for the scientific community. Thanks to your suggestions the paper, in our opinion, is notably improved. 

I only have a few comments.

Line 187: You should include a period after spp.

Response: Many thanks, we add it

Lines 175 – 185:  Remove bold formatting.

Response: Thanks we remove it.

Line 285: According to the WHO and CDC (references 8 and 9), there were 15 human cases in 2021.

Response: Many thanks for this comment. We add in the text this sentence: "There is a small discrepancy according to WHO and CDC data, which report 15 and 14 Guinea Worm cases, respectively, in 2021, but still down from the previous year. While in 2022, there were only four cases as of July 31, all in Chad.

Line 286: There are two periods after [49-50].

Response: Thanks, we remove it.

Figure 1: GWD human cases in 2021 instead of 2020.

Response: Thanks, we modify it.

Figure 1: In 2021, according to reference 9 (WHO) the number of cases were 15 instead of 14. Meanwhile, reference 8 (CDC) reported 14 cases. Please double-check and change the number of cases or the reference in Figure 1 and Line 88.

Response: Many thanks for this comment. We coherently modify the figure and  add this in the text :

“According to WHO and CDC data, 15 Guinea Worm cases were reported in 2021.”

While in the Figure. 1 modify as follow:“ Figure 1 shows the GWD human cases in African countries in 2021”.

Reviewer 2 Report (Previous Reviewer 2)

The authors have included most of the suggested corrections but there are still minor corrections

1. In table I column the nomenclature has not been corrected: Wuchereria NOT Wucheria : Brugia malayi NOT malay; Onchocerca, NOT Oncocerca

2. In table I, column 3, genders must be in italics.

3. In figure 1 Ethipoia should be replaced by Ethiopia

4. References 18 and 33 are identical

Author Response

REVIEWER 2

The authors have included most of the suggested corrections but there are still minor corrections

  1. In table I column the nomenclature has not been corrected:WuchereriaNOT Wucheria : Brugia malayi NOT malayOnchocerca, NOT Oncocerca
  2. In table I, column 3, genders must be in italics.
  3. In figure 1 Ethipoia should be replaced by Ethiopia
  4. References 18 and 33 are identical

Response: We thank you very much for the encouraging feedback on our manuscript. We followed your suggestions and believe that now the paper is more usable for the scientific community. Thanks to your suggestions the paper, in our opinion, is notably improved. Furthermore, a native English speaker revised the paper.  

Here our response:

  1. Many thanks, we modify according to your comment.
  2. Thanks we modify it
  3. Thanks me modify it
  4. Thanks we check and modify it.

Reviewer 3 Report (Previous Reviewer 3)

The authors have not revised the paper as recommended. PRISMA guidelines have not been followed in the revised manuscript also, as per the journal requirement/guidelines. The contents also are apparently unchanged.

This paper was submitted earlier also and rejected on the following grounds :

1.      Authors are advised to read the instructions to the authors carefully to understand how a manuscript intend to be published should be prepared .

2.      The paper does not qualify to be published as a review article  as the contents have neither been prepared nor structured like a review.

3.      As per the journal requirement a review is expected to provide concise and precise updates on the latest progress made in a given area of research and the systematic reviews should follow the PRISMA guidelines.

4.      This is an immature attempt to publish the knowledge available on the disease collected and presented in the paper in a haphazard manner and neither fits in the category of a review nor has the merit of being called a research article. Information copied from various sources has been included without even quoting the source of information.

Author Response

REVIEWER 3

The authors have not revised the paper as recommended. PRISMA guidelines have not been followed in the revised manuscript also, as per the journal requirement/guidelines. The contents also are apparently unchanged. 

This paper was submitted earlier also and rejected on the following grounds :

  1. Authors are advised to read the instructions to the authors carefully to understand how a manuscript intend to be published should be prepared .
  2. The paper does not qualify to be published as a review article  as the contents have neither been prepared nor structured like a review. 
  3. As per the journal requirement a review is expected to provide concise and precise updates on the latest progress made in a given area of research and the systematic reviews should follow the PRISMA guidelines.    
  4. This is an immature attempt to publish the knowledge available on the disease collected and presented in the paper in a haphazard manner and neither fits in the category of a review nor has the merit of being called a research article. Information copied from various sources has been included without even quoting the source of information.

Response: Reviewer 3 offends the authors. In authorship are the Directors of the Italian National Institute of Infectious Diseases (Emanuele Nicastri), directors of infectious and tropical disease University of Bari (Annalisa Saracino) clinics and young researchers.  It is a paper that aims to give a state of the art knowledge on the Guinea Worm. Collaboration with reviewers 1 and 2 led to a significant improvement of the paper and their suggestions were consistent and helpful to the paper. Yours are judgements not as a reviewer but as a judge and this is very misplaced.

I am sorry for the reviewer, but it deserves no further response.

Response 1: You can find our scientific paper production by searching for us on Scopus or Pubmed. Rather, I suggest you read this, https://febs.onlinelibrary.wiley.com/doi/full/10.1111/febs.15705, and remember that you are a reviewer, not a judge giving judgements.

Response 2: It is a literature review, we deal with tropical and neglected diseases and we believe this is the most appropriate format for Guinea Worm.

Response 3: 
This is not a systematic review but a reading review, and that is why there is no prism. If you search for our work, you will find meta-analysis, umbrella review, and systematic review. On Guinea worm, we chose to write an article that would put the state of the art of the disease and its eradication.

Response4: I completely disagree. By writing this, you offend the authors' scientific CVs. I think your way of reviewing was just a waste of time. You did not even deign to give a comment that would improve the paper, but just used an unacceptable words

This manuscript is a resubmission of an earlier submission. The following is a list of the peer review reports and author responses from that submission.

Round 1

Reviewer 1 Report

The manuscript corresponds to a review that summarizes the current knowledge of Guinea Worm Disease (also known as Dracunculiasis). The manuscript is well organized and comprehensively described. The references used are adequate. However, I have some minor and major comments.

Minor comments:

Line 44: Dracunculus medinensis is not well written, also it is not in italic. 

Line 67: Add a period at the end of Copepod [tiab]).

Line 68: between diagnosis and therapy there are two spaces.

Line 100: In humans,…

Line 104: impregnation?

Line 122: Replace are for is.

Line 140: Clinical

Line 161: between GWD and Among there are two spaces.

Line 175: What does mc mean?

Figure 2: The figure has low resolution. The information presented in the figure has been previously published, so Adapted from… should be included.

Table 2: Remove the row Clinical manifestation.

Line 191: Remove the period after larvae.

Line 192: With starts in lowercase. Remove the space between microscope and the period.

Line195: Missing period after ssp.

Line 199: Onchocerca should be in italic.

Line 227: D. medinensis should be in italic.

Line 254: Remove the space between GWEP and [42].

Line 256: 20 states from where?

Line 260: Add a space between measures and [43].

Line 269: Add a space between river and [45].

Line 302: According to the WHO there were 15 human cases

Lines 302 – 305: Missing reference

Major comments:

Table 1. I do not understand why you are presenting this table. The review is related to D. medinensis, so why are you talking about filaria parasites? If you want to keep the table, you should explain the relation between D. medinensis and filaria parasites. Also, the species should be written in italic.

Figure 1. Update the figure to 2021

Table 2: Include the references

Lines 269 – 275: missing reference

Why do you discard some references? For example, Hopkins et al 2021, 2020, 2019, 2018, 2017 (Progress toward global eradication), Darkase et al. 2021, Galan-Puchades 2019. These references should be included.

Please double-check the references, you are citing some references wrong. I mean, when I looked for the information in the papers that you are citing, sometimes I did not find it.

Related to author contributions, is not clear to me. How can be 26 authors in a review? What is the meaning of validation in a review that does not correspond to a systematic review? Why did not the corresponding author contribute to reviewing and editing the manuscript?

Reviewer 2 Report

The revised manuscript is an update on a classic neglected tropical disease. It is a complete, concise, and well-structured work that summarizes the previous aspects of this disease and clearly outlines the difficulties of its eradication.

Minor problems

 TEXT

1. The nomenclature of parasites should be in italics throughout the text.

2. In English, the term “pathophysiology” is more common than “physiopathology” (lane 68).

3. Correct “Cliical” on line 140

4. On several occasions Dracunculiasis spp (in italics) is cited, when it should be Dracunculus (in italics) spp (lanes 196 and 201 respectively)

5. Onchocerca (lane 197), on the contrary, should be in italics and onchocerchiasis (lane 197) in normal text.

6. Replace “dracunculiasiss” in lane 91 with “dracunculiasis”.

7. Unify upper and lower case letters in the denomination of “Guinea worm” (i.e. page 6 lane 166 and lane 170).

8. Replace mc with µm ( page 6 lane 173).

9. Throughout the text, replace spp in italics with spp .

10. Correct, substituting “.” by “,” on lines 191-192 on page 7.

11. Replace “infectiosin” on page 9 lane 265 with “infections in”.

TABLES

 Table 1.

    • General:

     - Nomenclature of parasites and vectors must be in italics.

     - Unify uppercase and lowercase letters mainly in columns 4 and 5

   • First lane: Substitute “localitation” for “location”

   • Lane 3: The correct name should be Brugia malayi and Brugia timori (in italics)

   • Lane 5 and 8 are identical. Delete (preferably 5)

Tab 2 ( page 6). Replace with Table 2

• Column 2. Unify uppercase and lowercase letters in the first word

• Line 2. Replace “lynfadenopathy” with “lymphadenopathy”.

FIGURES

Figure 1. Eliminate “human cases” in each country; it is already in the title of the figure

Figure 2: It is very illustrative but aesthetically a "collage" of different styles. I think the paper would improve a version with a univocal style (realistic or schematic)

REFERENCES

• I think it is not necessary to include DOI and PMID in each article.

• I think it is superfluous to add the Epub reference if the article has already been published.

• Evaluate the inclusion of a recent and pertinent article: Dracunculiasis: water-borne anthroponosis vs. food-borne zoonoses. Galán-Puchades MT.J Helminthol. 2019 Aug 22;94:e76.

Reviewer 3 Report

The paper does not qualify to be published as a review article as the contents have neither been prepared nor structured like a review.

Authors are advised to read the instructions to the authors carefully to understand how a manuscript intend to be published should be prepared.

As per the journal requirement, a review is expected to  provide concise and precise updates on the latest progress made in a given area of research and the systematic reviews should follow the PRISMA guidelines.

This is an immature attempt to publish the knowledge available on the disease collected and presented in the paper in a haphazard manner and neither fits in the category of a review nor has the merit of being called a research article. Information copied from various sources has been included without even quoting the source of information.
